# The Role of Autophagy in Vascular Endothelial Cell Health and Physiology

**DOI:** 10.3390/cells13100825

**Published:** 2024-05-11

**Authors:** Meghan Hu, Joseph M. Ladowski, He Xu

**Affiliations:** Transplant and Immunobiology Research, Department of Surgery, Duke University, Durham, NC 27710, USA; meghan.hu@duke.edu

**Keywords:** autophagy, mitophagy, endothelial cells, vascular disease, inflammation, atherosclerosis, therapeutics

## Abstract

Autophagy is a highly conserved cellular recycling process which enables eukaryotes to maintain both cellular and overall homeostasis through the catabolic breakdown of intracellular components or the selective degradation of damaged organelles. In recent years, the importance of autophagy in vascular endothelial cells (ECs) has been increasingly recognized, and numerous studies have linked the dysregulation of autophagy to the development of endothelial dysfunction and vascular disease. Here, we provide an overview of the molecular mechanisms underlying autophagy in ECs and our current understanding of the roles of autophagy in vascular biology and review the implications of dysregulated autophagy for vascular disease. Finally, we summarize the current state of the research on compounds to modulate autophagy in ECs and identify challenges for their translation into clinical use.

## 1. Introduction

### 1.1. Importance of Endothelial Cells in Vascular Health

The vascular endothelium is a single cell layer that lines the walls of all blood vessels. It performs a variety of important functions in not just the circulatory system but in multiorgan health and overall homeostasis. Endothelial cells (ECs) have a critical role in the physiologic regulation of vascular tone, permeability and resource exchange, angiogenesis, and hemostasis and also perform important anti-oxidant and anti-inflammatory functions [1]. EC dysfunction has been shown to play a significant role in the pathogenesis of atherosclerosis, hypertension, myocardial infarction, and other vascular diseases [2]. EC dysfunction can be induced by a variety of damaging risk factors, including chronic inflammatory disease, oxidative stress, and shear stress [2]. As such, reversing EC dysfunction and maintaining EC homeostasis has become an important target of research.

### 1.2. Brief Overview of Autophagy

Three primary types of autophagy have been defined in mammalian cells: (1) microautophagy, (2) chaperone-mediated autophagy, and (3) macroautophagy [3]. Microautophagy involves the direct invagination of the lysosomal membrane to engulf cytoplasmic cargo and is important in maintaining organelle size in times of nitrogen starvation, as well as in membrane homeostasis [3,4]. In chaperone-mediated autophagy, chaperone proteins identify cytoplasmic components with specific peptide motifs, which are then unfolded and translocated through the lysosomal membrane for degradation [4,5]. The final subtype, macroautophagy (which will be referred to simply as “autophagy”), is the best studied and is thought to be the major subtype in the cell [6,7]. It occurs at a low level basally but can be further upregulated during stress conditions such as nutrient or energy starvation to degrade either cytoplasmic materials or organelles into metabolites for biosynthesis or energy production [3]. 

Mitophagy is a special subclass of autophagy which selectively targets and transports superfluous or damaged mitochondria to the lysosome for degradation [8,9]. Unlike general autophagy, which is nonspecific, mitophagy requires the activation of specialized signaling pathways and specialized receptor proteins on the mitochondrial surface. Impairments in this process lead to the accumulation of defective mitochondria, leading to increased ROS and cell and tissue damage [10]. 

### 1.3. Understanding the Interplay between Autophagy and Endothelial Cells

Recent research has suggested that autophagy plays important roles in the cardiovascular system in order to support ECs’ response to environmental changes such as starvation, hypoxia, and ischemia [11]. Dysregulation of the autophagy system has been shown to disrupt EC function and turnover, leading to the accumulation of damaged cells and contributing to the pathogenesis of vascular disease [11]. A better understanding of the interplay between autophagy and ECs is crucial to deepen our knowledge of the mechanisms that contribute to EC dysfunction and vascular disease. Furthermore, understanding how autophagy functions in ECs will allow for improved therapeutic target design in the future treatment of these diseases.

In this review, we will summarize the molecular mechanisms of autophagy, the roles of autophagy (with a special focus on selective mitophagy) in endothelial self-repair, and what alterations in the autophagic process are associated with the development of endothelial dysfunction and vascular disease. Lastly, we will summarize the current experimental approaches to and methodologies of modulating autophagy in vitro and the potential clinical applications of autophagy regulators as new therapeutic strategies in precision medicine.

## 2. Molecular Mechanisms of Autophagy

In mammalian cells, autophagy is controlled both through upstream regulators (such as the MTORC pathway, among others) and the sequential activation of a panel of downstream autophagy-related gene (ATG) proteins [12]. It can be induced by cellular or environmental stimuli, such as nutrient deprivation, metabolic stress, or oxidative stress (further discussed later). The basic process of autophagy consists of several steps: initiation (involving the ULK1/ATG13/FIP200/ATG101 complex), nucleation (PI3K/Beclin1/VPS34/VPS15/NRBF2), elongation (ATG5/ATG12/ATG7/ATG10/ATG16L), autophagosome maturation (LC3/ATG3/ATG7/ATG4), fusion with the lysosome (involving Rab GTPases and SNAREs), and finally degradation (Figure 1).

### 2.1. Autophagy Initiation and Regulation

Unlike other forms of intracellular vesicle-mediated trafficking, the sequestering vesicles involved in autophagy do not bud off the membranes of pre-existing organelles already containing cargo [14]. Rather, autophagy is initiated by the de novo formation of an expanding membrane, known as a phagophore or isolation membrane, which sequesters cytoplasmic components [3]. 

Initiation is controlled by two major protein complexes. The Unc-51-like kinase (ULK) complex is composed of ULK1, ATG13, FIP200, and ATG101 and is an initiator kinase complex that recruits downstream factors and regulates their function via the phosphorylation of serine or threonine residues [15,16,17,18]. Upon stimulation, multiple copies of the ULK complex assemble to form a platform that facilitates the recruitment of other ATG proteins (as described later) and intermolecular autophosphorylation, resulting in the activation of kinase activity [19,20].

The phosphorylated ULK complex subsequently activates the second major protein complex involved in phagophore formation, the class III PI3K complex [21,22]. This is a large complex that includes vesicular sorting protein 34 (VPS34), VPS15, Beclin1, ATG14L, and NRBF2 [23,24,25]. The activated complex is responsible for the nucleation of the initial phagophore membrane, and its generation of phosphatidylinositol-3-phosphate (PI3P) is responsible for recruiting factors crucial to the formation of the autophagosome. Importantly, this complex recruits two interconnected ubiquitin-like (Ubl) conjugation systems, ATG8 and ATG12, to the phagophore, which are responsible for phagophore maturation [3,23,24].

### 2.2. Autophagosome Formation and Maturation

The phagophore engulfs organelles or cytoplasmic components to form a double-walled sequestering vesicle known as the autophagosome. The process of membrane expansion is mediated by two Ubl conjugations, the first of which involves the formation of the ATG5-ATG12-ATG16L complex by ATG7 and ATG10 [23,26]. ATG7 is also involved in the second Ubl, which additionally includes the ATG8 family proteins (LC3 and GABARAP), ATG3, ATG4, and the target lipid phosphatidylethanolamine (PE) [15,23].

In the final stage of autophagosome formation, the expanding phagophore must mature and close to form a vesicle, which traffics to and fuses with either an endosome or a lysosome. Currently, this stage is poorly understood, but some studies have suggested that LC3-II, a lipoprotein internal to the autophagosome, may also be involved in membrane closure [26]. The fusion of the autophagosome and lysosome is also not completely understood but is thought to be catalyzed by SNAREs [27]. Lastly, as the autophagosome fuses with the lysosome, its inner membrane is dissolved and degraded via hydrolases together with its sequestered materials in an acidic autophagolysosome [28].

### 2.3. Cargo Recognition and Selective Autophagy

Autophagy was initially considered to be a non-selective process in which bulk portions of the cytoplasm are sequestered for degradation during times of starvation. It has since been discovered that autophagy can also be a selective process to maintain the number and integrity of cellular organelles, protecting the cell from damage. 

Selective autophagy is mediated by selective autophagy receptors (SARs), which act as a bridge between the phagophore and the specific cargo [29]. SARs are able to recruit their cargo to the developing phagophore via the binding sites on the phagophore. Once the autophagosome is matured, they are degraded, along with their cargo [29]. In addition to their phagophore-binding sites, SARs also contain cargo-binding domains. These are frequently ubiquitin-binding domains, as autophagic cargo is often ubiquitinated [30,31,32,33]. However, SARs may also possess ubiquitin-independent cargo-binding domains, which may involve either direct binding to the cargo or the recognition of certain sugars or lipids which may be exposed on the cargo [34]. 

### 2.4. Molecular Players in Selective Mitophagy

There are multiple cellular mitophagy mechanisms, each of which can be promoted by various signaling cascades in different cellular contexts. One of the best studied mitophagy pathways is the phosphatase and tensin homologue (PTEN)-induced putative kinase 1 (PINK1)–Parkin pathway, which regulates ubiquitin-dependent mitophagy [35]. 

In normal mitochondria, PINK1 is transported to the inner mitochondrial membrane, where it is cleaved by proteases and subsequently degraded by the ubiquitin–proteasome system [36]. Following mitochondrial damage and membrane potential dissipation, PINK1 becomes stabilized on the outer mitochondrial membrane and activated through auto-phosphorylation, promoting Parkin recruitment [37]. Parkin then poly-ubiquinates other outer membrane proteins, which are subsequently also phosphorylated by PINK1 [38]. These phosphorylated poly-Ub chains are recognized by ubiquitin-dependent mitophagic SARs, such as p62, optineurin (OPTN), and nuclear dot protein 52 (NDP52), and anchor the mitochondria to autophagosomes through binding with LC3 [38]. 

Besides ubiquitin-dependent mitophagy, several mitochondrial proteins can also directly serve as SARs and interact directly with LC3 and GABARAP to mediate mitochondrial elimination [39]. Several inner membrane proteins, such as prohibitins and cardiolipin, have also been identified as SARs that are able to promote mitophagic removal via interactions with LC3 during times of energetic stress following their translocation to the outer membrane [30,40].

### 2.5. Noncanonical Autophagy

In addition to the canonical pathway controlled by ATG and associated proteins, evidence suggests that alternative pathways which do not require the full panel of ATGs can also result in autophagosome formation. For example, various Beclin1-independent forms of autophagy have been reported following cell treatment with pro-apoptotic compounds [41,42,43], cell differentiation [44], and bacterial toxin uptake [45]. In response to ammonia or glucose deprivation, noncanonical forms of autophagy which bypass ULK1 and/or the AMPK-mTORC1-ULK1 master regulator circuit (described later) have also been reported [46,47,48,49].

Interestingly, noncanonical autophagy may represent a tie between autophagy and the immune system. Stimulator of interferon genes (STING) plays an important role in innate immunity and the inflammatory response by inducing interferons and other cytokines in response to the activation of the cyclic GMP-AMP synthase (cGAS) system by infection or tissue damage [50]. Studies have shown that in addition to its immune functions, STING is able to trigger noncanonical autophagy by inducing LC3 lipidation in a pathway that is ULK- or VPS34/Beclin-independent (rather, it appears to be dependent on WIPI2 and ATG5) [51,52,53]. Additionally, STING is able to control energy-stress-induced autophagy and energy metabolism through the regulation of autophagic SNAREs [54]. Finally, studies have found that STING is also able to trigger an additional form of noncanonical mitophagy by sensing mtDNA and triggering STING-dependent mitophagy in endothelial cells via interferon gamma-inducible factor 16 (IFI16) and NFkB signaling [55]. Importantly, mtDNA also triggers a STING-dependent immune response in ECs, establishing a further link between non-canonical autophagy and the inflammatory response [55]. Taken together, these findings suggest the possible role of autophagy in modulating the innate immune response. 

## 3. Autophagy in Vascular Endothelial Cell Function

### 3.1. Role of Autophagy in Endothelial Cell Survival 

Vascular endothelial cells have a crucial role in maintaining normal nutrient and oxygen exchange, vascular homeostasis, and the normal function of the cardiovascular system [56]. Increasing evidence suggests that autophagy plays an important role in promoting vascular endothelial cell survival under both physiologic and pathologic conditions [57]. Research suggests that autophagy enables vascular ECs to adapt plastically to physiologic shear stress [58]. Conversely, insufficient autophagy impairs endothelial cell alignment in response to changes in blood flow and causes increased EC apoptosis [59]. Impairments in autophagy have been shown to induce and aggravate vascular EC morphological abnormalities and lead to increased apoptosis [60,61].

Autophagy also protects ECs against damage from oxidative stress, which promotes cell survival. For example, Rezabaksh et al. found that human umbilical vein endothelial cells (HUVECs) that were exposed to 30mM of glucose displayed decreased cell viability and increased intracellular ROS. HUVECs treated with both 30 mM of glucose and 100nM of rapamycin (which upregulates autophagy) demonstrated improved cell health and decreased intracellular ROS [62]. This protective effect of autophagy against oxidative stress may be mediated through selective mitophagy, as studies have found that the inhibition of autophagic flux by chloroquine or bafilomycin A1 increases mitochondrial ROS, while restoration of autophagic flux leads to reduced mitochondrial ROS accumulation and improved cellular function [63]. Furthermore, in cells exposed to a high-fat diet, downregulating mitophagy through ATG5 deletion leads to EC injury (as demonstrated by severe podocyte injury, impaired endothelial fenestrae, and compromise of the endothelial glycocalyx) and increased apoptosis [64]. Conversely, upregulating mitophagy in cells exposed to hyperglycemia-induced oxidative injury has a protective effect and decreases mitochondrial ROS and endothelial cell apoptosis [64].

While autophagy is usually thought to play a protective role in preserving EC function, overactivation can have a damaging effect and lead to cell death [65]. For example, while moderate mitophagy selectively eliminates damaged organelles and can lead to increased ATP production [63,66], overactivation of mitophagy leads to cell death due to no energy supply [67]. The degree to which mitophagy (and other autophagic processes) is activated appears to be influenced by the intensity and duration of stress. The regulatory mechanisms of autophagy in ECs as well as the interplay between autophagy and apoptosis will be discussed later; however, this remains an ongoing area of research, and its precise role and the mechanisms according to which the overactivation of autophagy can lead to cell death are not fully understood. 

### 3.2. Additional Roles of Autophagy in Vascular Function

Nitric oxide (NO) plays many important roles in the vasculature, including the regulation of vascular tone and blood flow, and studies have found a role of autophagy in the regulation of nitric oxide bioavailability. Inhibiting autophagy in ECs via ATG3 knockdown impairs the phosphorylation of endothelial nitric oxide synthase (eNOS) and reduces NO production in response to shear stress [68]. Activating autophagy under steady laminar shear stress in ECs increases eNOS expression and improves vascular tone [69]. Autophagy in ECs may also help in the regulation of vascular permeability. ECs require autophagy to regulate tight junction proteins to maintain endothelial barrier integrity, and the inhibition of autophagy in ECs increases vascular permeability via an ROS-dependent mechanism [70]. 

Finally, autophagy may also be involved in ECs’ secretion of von Willebrand factor (vWF), a glycoprotein crucial to platelet adhesion in hemostasis. Studies have shown that vWF localizes near the autophagosomes in ECs and that the knockdown or deletion of ATG5 or ATG7 impairs ECs’ secretion of vWF and alters hemostasis [71]. Pharmacological inhibition of autophagy using chloroquine treatment has also been shown to result in downregulated vWF secretion [72].

### 3.3. Autophagy-Mediated Regulation of Angiogenesis

Research has also explored the role of autophagy in angiogenesis, the growth of new blood vessels from the existing vasculature. Angiogenesis is usually constitutively inhibited but can be triggered by environmental factors such as nutrient deprivation, hypoxia, ischemia, or blood flow variation [73]. Notably, these are many of the same stressors that also upregulate autophagic flux [74]. 

Some studies have shown that pharmacologic inhibition of autophagy increases angiogenesis in an eNOS-dependent manner, suggesting that its regulation of NO bioavailability may be anti-angiogenic [75]. In human umbilical vein endothelial cells, pharmacologic upregulation of autophagy with magnolol led to decreased migration and tube formation [76]. Other studies have found that autophagy is responsible for vascular endothelial growth factor receptor-2 degradation and impaired angiogenesis in in vitro models of diabetes and that excessive autophagy is responsible for the abrogation of angiogenesis in mesenchymal stem cells exposed to the sera of patients with diabetes [62,77]. 

However, many other studies have also demonstrated the pro-angiogenic role of autophagy. One of the most well-known pro-angiogenic growth factors is vascular endothelial growth factor (VEGF) [78]. Work by Spengler et al. confirmed the important role of autophagy in the induction of functional angiogenesis by demonstrating that VEGF induced the phosphorylation of ULK1 via AMPK activation in human umbilical vascular endothelial cells [79]. In bovine aortic endothelial cells, the induction of autophagy via ATG5 overexpression was found to increase tube formation and cellular migration by activating VEGF. The inhibition of autophagy in these cells impaired VEGF-induced angiogenesis [80,81]. In mice, ATG5 deficiency also leads to impaired VEGF- and flow-mediated neoangiogenesis in both microvascular and macrovascular beds [82]. Further work is needed to elucidate autophagy’s exact function in vessel development.

## 4. Regulation of Autophagy in Endothelial Cells

### 4.1. Environmental Regulation of Autophagy

Endothelial cell damage can be induced either by external mechanical forces, such as shear stress, mechanical stretching, or vascular wall stress, or by regulated cell death [2]. As previously discussed, autophagy under these physiologic stress conditions is thought to help cells adapt to various stressors and support normal EC function and homeostasis.

Shear stress is the mechanical force caused by the sliding of blood on the surface of the endothelium and plays important roles in modulating the biological functions of ECs, including gene expression, proliferation, migration, morphogenesis, permeability, thrombogenicity, and inflammation [83,84]. It is thought that the unidirectional high-magnitude wall shear stress produced by laminar blood flow in straight vessels is vascular-protective. In contrast, the shear stress associated with disturbed blood flow (low or oscillatory shear stress in areas with branches or curves) is pro-atherogenic [83,84]. Recent work has shown that autophagic response in ECs is regulated by shear stress [58,68,85,86]. Laminar shear stress has been demonstrated to increase EC autophagy, as shown by LC3-II formation and the upregulation of autophagic genes [86]. However, studies examining the effect of low or oscillatory shear stress on EC autophagy have had mixed results. Yang et al. found that the levels of Beclin1 and LC3-II were lower under low-shear-stress conditions compared to “physiological” shear stress and that the impaired autophagic function under low shear stress was associated with vascular dysfunction [87]. Other authors have found that low shear stress enhanced EC autophagy with increased LC3-II [88]. The reason for these discrepancies is not yet clear but could be related to differences in the experimental settings. 

Finally, autophagy in ECs is also modulated by exposure to cardiovascular risk factors. Treatment of ECs with oxidized low-density lipoprotein (ox-LDL) triggers autophagic responses [89,90]. Increased autophagic flux in ECs accelerates the catabolism and clearance of lipid substrates, which may prevent lipid-overload-induced endothelial toxicity [90,91]. Advanced glycation end products (AGEs), which are present in diabetes mellitus, have also been shown to increase autophagy [92] via the upregulation of ROS production (Figure 2). 

### 4.2. Signaling Pathways Modulating Autophagy in Endothelial Cells

Autophagy is a tightly regulated process in ECs, as insufficient autophagy can be deleterious but excessive levels may also be harmful. Autophagosome biogenesis is induced by a wide variety of signals, including nutrient shortage, cellular stress (oxidative stress, hypoxia, etc.), and the emergence of aberrant protein aggregates or damaged organelles [93]. While a review of all of these pathways is beyond the scope of this paper, transcription factor EB (TFEB) and its associated proteins, the class O of the forkhead box transcription factor (FOXO) family), the sirtuin family of enzymes, the tumor suppressor p53, and the transcription factors E2F1 and NFkB are all implicated in autophagosome initiation. Many of these signals converge at the mTOR signaling pathway, which is a master regulator of cellular metabolism and promotes cell growth in response to environmental cues [94].

### 4.3. The mTORC Pathway

mTOR forms two distinct signaling complexes, mTOR complex 1 (mTORC1) and mTORC2, through binding with different companion proteins. These two kinase complexes have specific substrate preferences and elicit distinct downstream signaling events to modulate cellular function [95]. mTORC1 promotes anabolic cellular metabolism, cell growth, and proliferation and blocks catabolic processes such as autophagy. Specifically, mTORC1 phosphorylates ULK complex subunits to block downstream events for autophagosome biogenesis [15,96,97]. When mTORC1 is attenuated in response to either nutrient shortage or pharmacological inhibition, autophagy is induced [95,97].

Hypoxia and the growth factor concentrations are also able to regulate autophagy at least in part through mTORC1, and hypoxia can inhibit mTORC1 and induce autophagy even with adequate nutrients and growth factors [98,99] In mammals, the AMP-activated protein kinase (AMPK) also plays a major role in autophagy induction in response to low glucose levels via mTORC1 inactivation, as well as the direct phosphorylation of ULK1 [100,101].

### 4.4. Epigenetic Regulation of Autophagy

In addition to regulation by intracellular signaling pathways, autophagy also undergoes epigenetic and post-translational regulation via DNA methylation, histone methylation and acetylation, microRNAs, ubiquitination, and phosphorylation [102]. While a full review of these regulatory mechanisms is beyond the scope of this review, it is interesting to note that some of these pathways have also been implicated in the development of vascular disease. For example, studies have found that in vascular ECs, in response to stimulation by angiotensin II, the deacetylation of FOXO3a activates autophagy and subsequently leads to vascular inflammation [103]. Methylation of autophagy promoters, such as the Beclin1 promoter, has also been implicated in impairments in autophagic flux in vascular ECs and the subsequent development of atherosclerotic lesions [104]. However, more research is needed to completely elucidate the role of epigenetics in autophagy regulation in vascular ECs.

### 4.5. Crosstalk between Endothelial Cell Autophagy and Apoptosis

In recent years, the complex crosstalk between autophagy and apoptosis has received increased attention. Autophagy often functions as a cryoprotective stress adaptation which inhibits apoptosis (especially in starvation settings). For example, BAX and BAK are two mitochondrial BCL2 family proteins required for apoptotic cell death. A study by Lum et al. showed that immortalized IL-3-dependent cells generated from the bone marrow of *Bax* and *Bak* double-knockout mice failed to undergo apoptosis after IL-3 withdrawal. In contrast to the WT cells, which immediately underwent apoptosis, these *Bax^−/−^Bak^−/−^* cells underwent a month-long autophagic process that caused a severe reduction in cell size with the removal of most of the cytoplasm [105]. However, inhibition of autophagy via *Atg5* or *Atg7* knockdown or pharmacologically using 3-methyladenine (3-MA) or chloroquine led to the death of the *Bax^−/−^Bak^−/−^* cells, confirming that autophagy had suppressed apoptosis in this setting [106]. As previously detailed, other studies have similarly shown that inactivating *Atg* genes or downregulating autophagy can lead to increased apoptosis [57,58,59,60,61].

Recent studies have also suggested that mitophagy may be particularly important for inhibiting apoptosis by clearing damaged mitochondria before they can induce cell death. For example, the pre-treatment of cells with rapamycin (an inducer of autophagy) protects cells from mitochondrial outer membrane permeabilization (MOMP)-dependent apoptotic stimuli while causing a decrease in the mitochondrial mass by ~50% [107]. It is interesting to note that the inhibition of autophagy does not always induce cell death through apoptosis. For example, malignant glioma cells usually exhibit a surge in autophagy following treatment with DNA-damaging agents. When autophagy is inhibited, the cells do not undergo apoptosis; instead, they undergo non-apoptotic cell death with micronucleation [108]. Furthermore, mouse epithelial cancer cells in which both apoptosis and autophagy are suppressed die from necrosis [109]. 

In other situations, autophagy can more directly lead to cell death when apoptosis is inhibited. Caspases are a family of proteases which are essential for apoptosis. In macrophages treated with LPS, pan-caspase inhibition with the compound Z-VAD-FMK leads to autophagy-mediated cell death, as confirmed by the presence of increased autophagosomes using electron microscopy. This was attenuated following Beclin1 knockdown, further supporting the role of autophagy here in leading to cell death [110]. Other studies have found that the autophagic cell death induced by pan-caspase inhibition is associated with autophagy-mediated depletion of the antioxidant enzyme catalase. This effect is abrogated following either antioxidant administration or autophagy inhibition through ATG7 or Beclin1 depletion, suggesting that the autophagic elimination of catalase leads to irreversible cellular damage and hence necrotic cell death due to the overaccumulation of ROS [111]. Interestingly, the authors also found that in the same system, nutrient-starvation-induced autophagy did not lead to catalase elimination or cell death, indicating that the autophagic targeting of catalase may depend on the upstream signals that cause autophagy [112]. 

Furthermore, some signals can trigger both apoptosis and autophagy. For example, ROS accumulation can trigger pro-apoptotic MOMP as well as stimulate ATG4 and upregulate autophagy [113]. Similarly, the sphingolipid ceramide is a potent inducer of apoptosis that acts via triggering MOMP but has also been shown to stimulate autophagy [114]. Another sphingolipid, sphingosine-1phosphate, inhibits ceramide-induced apoptosis but also induces autophagy [115]. In addition, increases the in free Ca^2+^ concentrations in the cytosol are a pro-apoptotic signal [116]. However, an increase in Ca^2+^ is also able to trigger autophagy through the downstream activation of AMPK, which then inhibits mTOR and de-inhibits autophagy [117]. Ca^2+^ furthermore stimulates calpains, which are also able to contribute to autophagy or apoptosis [118,119]. Taken together, these findings suggest that the intermingled apoptotic and autophagic pathways may be quite sensitive to intracellular signals and that slight changes in the balance of intracellular compounds may push the cell one way versus another. 

Other molecular connections in the signaling pathways that regulate both processes include p53 (induces apoptosis but also induces autophagy), the PI3K/Akt pathway (inhibits apoptosis but also inhibits autophagy), BCL family proteins (which interact and modulate both the apoptotic and autophagic pathways), FAS-associated death domain proteins (induce both apoptosis and autophagic cell death), and some ATG proteins, which directly regulate both processes [120]. 

## 5. Autophagy Dysregulation in Endothelial Dysfunction

As autophagy is involved in the regulation of many crucial functions of the vascular system, its impairment is associated with a wide range of vascular diseases, including atherosclerosis, vascular ischemic disease and ischemia/reperfusion injury, and pathological angiogenesis (Table 1).

### 5.1. Atherosclerosis

Endothelial dysfunction is thought to be the first step in the pathogenesis of atherosclerosis [135]. Decreased EC autophagy contributes to senescence and apoptosis, which leads to endothelial dysfunction [136]. Insufficient EC autophagy also increases the levels of vWF, p-selectin, vascular cell adhesion molecule-1 (VCAM-1), and intercellular adhesion molecule-1 (ICAM-1), the accumulation of which increases the risk of arterial thrombosis, as well as promoting the vascular infiltration of foam cells and macrophages, thus contributing to the development of atherosclerotic lesions [121,122]. 

Defects in functional autophagy in ECs have also been shown to accelerate the pathogenesis of atherosclerosis in several in vivo studies. Torisu et al. demonstrated that *ApoE^−/−^* mice with an endothelial specific *Atg7* deletion had an increased plaque burden after 12 weeks of high fat diet and increased lipid deposition throughout the entire aorta as compared to *ApoE^−/−^* mice with intact autophagy [123]. Vion et al. additionally studied the effect of insufficient EC autophagy on atherosclerosis under physiological and pathological shear stress (high and low shear stress, respectively) in *ApoE^−/−^* mice with an endothelial-specific *Atg5* KO fed a high-fat diet. They found that the *Atg5* KO mice developed massive plaques in the descending aorta, an area that is normally atheroresistant, compared to the controls; the plaque burden was similar in both groups in the atheroprone regions of the aorta [59]. This suggests that EC autophagy is likely already impaired in these atheroprone regions due to low shear stress, while further defects in EC autophagy promote the development of atherosclerosis in previously resistant regions [59]. 

Impairments in endothelial cell lipophagy, the autophagic breakdown of lipids, may also contribute to atherosclerosis. Lipid accumulation in the arterial walls is a key event in the development of atherosclerosis [137]. Thus, lipophagy is a way for endothelial cells to regulate lipid accumulation and maintain homeostasis. In cultured endothelial cells, it has been shown that exposure to ox-LDL results in an increased number of ox-LDL-containing autophagosomes [89,90,138,139]. However, overexposure to ox-LDL or deficits in lipophagy can lead to cellular damage. Studies have shown that HUVECs treated with prolonged exposure to oxidized LDL had decreased lipophagy and subsequently increased lipid accumulation [124]. In mice, inhibition of autophagic flux via the transient knockdown of ATG7 has been shown to result in increased intracellular I-LDL and ox-LDL [123]. Other authors have found that inhibition of autophagy in cholesterol-loaded vascular smooth muscle cells led to significantly increased cell death, while rapamycin-induced upregulation of autophagy protected cells from cholesterol-overload-induced cell death [125]. While further work must be carried out to confirm whether this effect is also true in vascular ECs, taken together, these findings suggest that deficient lipophagy may be key to the development of atherosclerotic lesions due to cholesterol overload. 

Mitophagy may also play an especially important role in the pathogenesis of atherosclerosis. Oxidative stress is considered a key player in the progression of inflammation and cellular damage, leading to the development of atherosclerotic lesions [140]. Mitophagy decreases oxidative damage by reducing endothelial mtROS overproduction and clearing damaged mitochondria. As such, defects in mitophagy have been shown to worsen atherosclerosis. Exposure to high glucose induces mtROS production and decreases mitophagy, which leads to dysfunctional mitochondria accumulation and the acceleration of endothelial injury [126]. However, other studies have suggested that overactivation of mitophagy may worsen atherosclerosis. In aortic ECs, for example, LDL-ox leads to the overexpression of NR4A1, which abnormally activates mitophagy, mediated by the PINK1–Parkin pathway [127]. These leads to dramatic decreases in mitochondrial supply and massive endothelial apoptosis due to insufficient energy supply, which worsens the progression of atherosclerosis [127]. Thus, while defects in autophagy and mitophagy have been shown to play a role in the pathogenesis of atherosclerosis, our understanding of their exact mechanism remains unclear.

### 5.2. Ischemic Disease and Ischemia/Reperfusion Injury

Studies on the role of autophagy in ischemic disease and in ischemia/reperfusion injury in ECs are limited, with mixed results. Some research has suggested the protective role of mitophagy in ECs in IRI [128,129]. However, other studies suggest the maladaptive role of mitophagy in IRI. In an in vitro model of microvascular IRI following percutaneous coronary intervention, treatment with melatonin protected ECs from damage by downregulating mitophagy, suggesting that excessive mitophagy in IRI may have a deleterious effect [130]. In rodent lung models of ischemia/reperfusion injury, the induction of mitophagy via treatment with rapamycin promoted apoptosis and inhibited the proliferation of pulmonary microvascular endothelial cells, aggravating lung injury [131]. Inhibition of mitophagy via treatment with 3-MA attenuated this effect. Thus, the exact role of endothelial cell autophagy and mitophagy in ischemic disease/IRI remains to be determined.

### 5.3. Pathological Angiogenesis

In recent years, a small but increasing amount of evidence has suggested that autophagy may play a role in the regulation of pathological angiogenesis, as occurs in solid tumors and in diseases of chronic inflammation, such as diabetic retinopathy. However, the findings in this area have also been mixed. Several studies have suggested that upregulating autophagy may play a protective role against aberrant neovascularization. For example, in an in vitro model of breast carcinoma, induction of mitophagy was able to attenuate tumor neovascularization [132]. In a mouse melanoma tumor model, defective autophagy due to the heterozygous disruption of Beclin1 led to accelerated tumor growth and a more pro-angiogenic phenotype under hypoxia as compared to that in the controls [133]. In human ECs, treatment with endostatin, a potent angiogenesis inhibitor, increased autophagic cell death, thus preventing vessel development [134]. 

Other studies have suggested the deleterious role of autophagy in pathological angiogenesis and that downregulating autophagy improves neovascularization. Studies have found that treatment with GRP, an inducer of tubule formation, was found to decrease the expression of the autophagy-related genes ATG5, Beclin1, and LC3 [141]. Other studies have found that in human glioblastoma xenografts, knockdown of *ATG7* combined with treatment with the VEGF-neutralizing antibody bevacizumab disrupted tumor growth and helped prevent tumor resistance to anti-angiogenic therapy [142]. Further studies have also shown that Atg5 depletion in ECs can exacerbate functional abnormalities in tumor vasculature, suggesting that autophagy may be important for vascular normalization in solid tumors [143].

Besides cancer, autophagy has also been implicated in other diseases involving abnormal neovascularization, such as diabetic retinopathy. In vitro studies in retinal endothelial cells have shown that hyperglycemia promotes angiogenesis, enhances ROS production, and induces autophagy [81]. However, pre-treatment with autophagy inhibitors improved cell viability and reduced tube formation and cell migration [144]. In a recent genetic study, Wang et al. examined a database of patients with diabetic retinopathy and identified 23 potential autophagy-related genes that were differentially regulated in diabetic retinopathy [145]. Thus, while the current state of the research has identified potential associations between autophagy and pathological angiogenesis, as demonstrated in solid tumors or in diabetic retinopathy, further work is needed to clarify the exact role of autophagy in this process. 

## 6. Clinical Relevance and Potential Therapeutic Interventions

In view of the important role played by autophagy in vascular diseases, there is an emerging interest in developing pharmacological regulators of autophagy and mitophagy to treat vascular pathologies (Table 2). However, the majority of the studies in this space remain pre-clinical, and human studies on autophagy regulators remain sparse.

The available data have shown that certain compounds such as pomegranate, *Ginkgo biloba* extract, and the synthetic chemical known as PMI are able to modulate mitophagy in vitro, although this has not been tested specifically in ECs [146,147,148]. However, quercetin, a flavonoid antioxidant, has been shown to protect human endothelial cells against the oxidative damage induced by high glucose exposure by upregulating autophagy [149]. 

Furthermore, a novel type of molecule that was originally developed for cancer treatment, known as the selective, ubiquitin-mediated autophagy-targeting chimera (AUTAC), is now attracting interest as a mitophagy-modulating therapeutic. In vitro studies have shown that AUTAC is able to augment mitochondrial turnover and facilitate the removal of damaged mitochondria in fibroblasts [150]. Although it has also not been tested in ECs, it may be a promising therapeutic target for treating vascular pathologies. 

### 6.1. mTORC Inhibitors

Rapamycin is an FDA-approved mTORC1 inhibitor widely used as an immunosuppressant following organ transplant and in cancer treatment. In rodent models of ischemic stroke, rapamycin was shown to significantly enhance mitophagy, inhibit mitochondrial dysfunction, and decrease infarct volume [151]. In rodent models of spinal IRI, rapamycin was similarly shown to upregulate mitophagy and attenuate IRI-induced apoptosis [171]. A derivative of rapamycin, everolimus, is also currently used as an immunosuppressant in transplant, targeted therapy for certain cancers, and in cardiovascular drug-eluting stents to inhibit restenosis. Studies have shown that everolimus inhibits mTORC1 with greater specificity than rapamycin, making it a promising target for mitophagy regulation [152]. Other related agents in this class include temsirolimus, ridaforolimus, and deforolimus, which are currently under clinical development [153,154]. 

### 6.2. SIRT1 Activators

Studies have shown that the drugs resveratrol and polydatin are able to exert cardioprotective effects and decrease myocardial IRI via mitophagy activation and mTORC inhibition [156,172]. Resveratrol in particular has been shown to be able to attenuate endothelial oxidative injury by upregulating autophagy through the activation of TFEB [155]. A related drug, salidroside, has been shown to protect against ox-LDL-induced endothelial injury by enhancing autophagy via the upregulation of the SIRT1 and FOXO pathways [157]. 

### 6.3. Melatonin

Melatonin has been shown to upregulate mitophagy in macrophages and ameliorate the progression of atherosclerosis in an atherogenic mouse model by activating the SIRT3-FOXO3–Parkin signaling pathway [158]. In other studies utilizing mouse models of cardiac IRI, melatonin was shown to protect against IRI by preventing excessive mitophagy and mitophagy-mediated cell death [130]. Thus, while melatonin may have some beneficial effects as a modulator of mitophagy, its mechanism of action is not fully understood. 

### 6.4. Metformin and Other Drugs

Metformin has been shown to promote mitophagy in patients with type 2 diabetes via AMPK activation and the upregulation of genes of such as PINK1, Parkin, NIX, and LC3 [159]. The downstream effects of this upregulated mitophagy are not yet well understood. Statins, such as atorvastatin and rosuvastatin, have long been used to treat cardiovascular disease, and in addition to their lipid-lowering properties, they have been shown to be able to induce autophagy as well [160,161]. Research on statin-induced autophagy in vascular ECs is very limited; however, a 2018 study by Dang et al. showed that atorvastatin is able to prevent cellular dysfunction in HUVECs exposed to angiotensin II [162]. Further work by Zhao et al. in 2019 found that varying doses of atorvastatin universally induced autophagy but exerted differential effects on HUVECs and that high doses may be used to suppress angiogenesis [163]. More work is needed to fully elucidate this effect. There has also been increasing interest in recent years in targeting de-ubiquinating enzymes to selectively increase mitophagy, even in the absence of Parkin and PINK1. The most promising of these is USP30, a protease located in the outer mitochondrial membrane that inhibits Parkin-mediated mitophagy [170]. 

### 6.5. Non-Pharmacologic Modulation of Autophagy

Research is also underway on non-pharmacological ways to modulate autophagy, such as exercise and caloric restriction. Studies have found that exercise training is able to upregulate autophagy, which may exert a cryoprotective role by limiting ROS and inflammation [164,165,166]. Other authors have found that in certain situations, exercise training can inhibit excessive autophagy [167,173,174]. Interestingly, it has also been reported that excessive exercise can also trigger too much autophagy and lead to deleterious effects [175]. Caloric restriction and intermittent fasting have also been explored as ways to upregulate autophagy and promote cardiovascular health. Some studies have suggested that intermittent fasting upregulates autophagy and protects myocardiocytes from IRI in mice [169]. Sun et al. further demonstrated that obese individuals who participated in a 7-day fasting intervention had increased autophagy, reduced levels of arterial injury markers, and improved endothelial progenitor cell function [168].However, as previously discussed, overactivation of autophagy due to excessive caloric restriction may have detrimental effects and trigger apoptosis. 

Overall, while the modulation of autophagy and mitophagy may be a promising strategy for targeting vascular disease, the number of potentially effective and specific therapeutics remains very limited, and further studies need to be carried out before they are able to be widely used in humans.

## 7. Conclusions

In recent years, a growing body of research has established the role of autophagy in the vascular endothelium, demonstrating that autophagy is an important mechanism by which ECs are able to adapt to stressors and changes in blood flow. It has also become clear that autophagy affects many crucial aspects of EC biology and vascular function, including permeability, secretion, and angiogenesis. While its exact role is not yet fully understood, dysregulation of autophagy in ECs has been implicated in the pathogenesis of a variety of vascular diseases. Thus, an in-depth understanding of the pathways and environmental factors that influence autophagy in ECs will be crucial to improving our knowledge of and ability to treat vascular pathologies. 

It has become clear that autophagy in ECs is highly clinically relevant, and in recent years, effort has turned towards the development of pharmacological agents aimed at modulating autophagy. Despite this clinical potential, no therapeutics that specifically target autophagy are currently available for use in humans. Several overarching challenges must be addressed before the modulation of autophagy is ready for clinical use. Firstly, our knowledge of the role of autophagy in vascular disease remains incomplete, and it will be important to precisely identify the pathologies that are caused or aggravated by alterations in autophagy versus those in which autophagy serves as a compensatory mechanism. Secondly, it is important to note that while it can be cryoprotective, excessive autophagy may also lead to cell death. To prevent adverse consequences, further work is needed to clarify the crosstalk between autophagy and apoptosis. Lastly, our current pharmacologic interventions have very limited specificity for the autophagic process. For example, while rapamycin can increase autophagy via mTORC1 inhibition, this process also inhibits cellular growth, proliferation, and translation. Furthermore, many parts of the autophagy machinery are also involved in other cellular processes, and modifying their function may have unintended consequences. Finally, there are not yet any autophagy modulators that are specific to a certain cell type, such as ECs, another factor which makes the clinical implementation of targeted therapy challenging. 

Autophagy in ECs is a clinically important and fast-growing area of research. Before it can be translated into the clinic for human use, additional work is needed to understand how the inhibition or activation of endothelial cell autophagy functions in the vascular system. However, modulation of autophagy has great potential as a strategy for the treatment of vascular pathologies and represents an exciting future direction for the development of new and more targeted therapeutics. 

## Figures and Tables

**Figure 1 cells-13-00825-f001:**
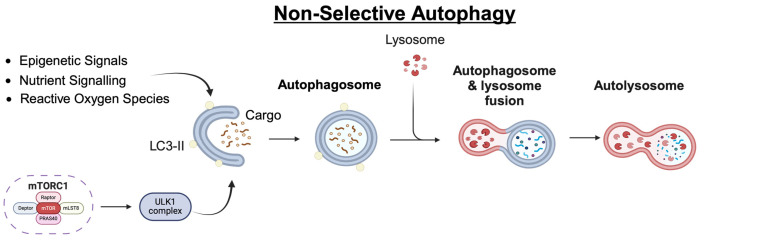
Overview of nonselective autophagy and mitophagy—a schematic depicting the major steps in the process of mitochondrial-focused macroautophagy and selective mitophagy. The figure was generated using BioRender.com and is modified from a template provided by BioRender.com and previously published by Hansen et al. [13].

**Figure 2 cells-13-00825-f002:**
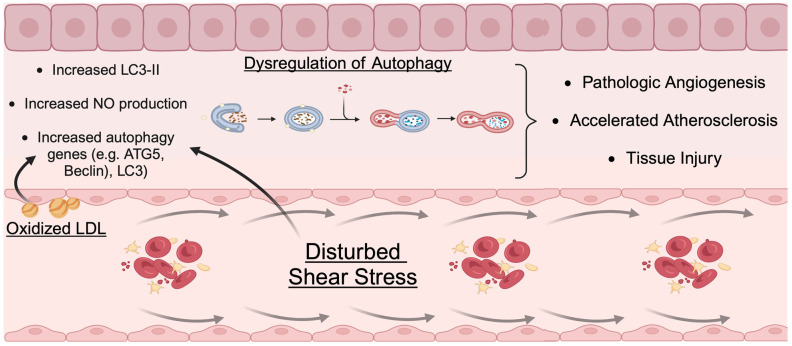
Mechanisms of external mechanical forces in inducing endothelial autophagy. Potential mechanisms of disturbed shear stress and oxidized low-density lipoprotein (LDL) resulting in dysregulation of vascular endothelial autophagy and upregulation of LC3-II, nitric oxide (NO) production, and autophagy-related genes. Overall, these factors are thought to be potential mechanisms leading to pathologic angiogenesis, accelerated atherosclerotic plaque development, and microvascular tissue injury.

**Table 1 cells-13-00825-t001:** Autophagy in vascular disease – select ways in which autophagy dysregulation has been implicated in the pathogenesis of vascular diseases.

Disease	Autophagy Level	Study Methods	Effect	Ref
Atherosclerosis	↓	Treated HUVECs with ox-LDL	Increased thrombogenicity via increased secretion of vWF and p-selectin	[121]
	↑	3PO induction of autophagy	Decreased plaque formation via decreased secretion of ICAM-1 and VCAM-1	[122]
	↓	Atg7 deletion in ApoE^−/−^ mice fed high-fat diet	Increased plaque burden, aortic lipid deposition	[123]
	↓	Atg5 KO in ApoE^−/−^ mice fed high-fat diet	Plaque development in previously atheroresistant areas	[59]
	↓	Prolonged exposure of HUVECs to ox-LDL	Increased lipid accumulation	[124]
	↓	Atg7 knockdown in HUVECS	Increased lipid accumulation	[123]
	↓	3MA inhibition of autophagy	Increased vascular smooth muscle cell apoptosis due to free cholesterol overload	[125]
	↑	Rapamycin induction of autophagy	Cryoprotective effect in vascular smooth muscle cells against apoptosis triggered by free cholesterol overload	[125]
	↓	Treated HUVECs with high glucose	Impaired mitophagy, increased mtROS production, dysfunctional mitochondria accumulation	[126]
	↑	Treated aortic ECs with ox-LDL	Overactivation of PINK–Parkin mitophagy, EC apoptosis	[127]
Ischemia/Reperfusion injury	↑	UA induction of mitophagy	Increased cell viability and proliferation and decreased mitochondrial damage	[128]
	↑	Rapamycin induction of autophagy	Decreased brain mitochondrial EC apoptosis following IRI, decreased ROS	[129]
	↓	Treatment with melatonin in a mouse model of IRI following PCI	Decreased cardiac microvascular EC damage, prevention of excessive mitophagy-mediated cell death	[130]
	↑	Rapamycin induction of autophagy in a mouse model of IRI following lung transplant	Increased apoptosis due to excessive mitophagy, inhibited proliferation of pulmonary microvascular ECs	[131]
Pathological angiogenesis	↑	Decorin induction of mitophagy in breast carcinoma model	Attenuated tumor neovascularization	[132]
	↓	Heterozygous disruption of Beclin1 in mouse melanoma model	Accelerated tumor growth, more pro-angiogenic phenotype under hypoxia	[133]
	↑	Endostatin induction of autophagy	Increased autophagic cell death and prevented vessel development	[134]
	↓	Atg7 knockdown and treatment with bevacizumab	Disruption of tumor growth and angiogenesis	[134]
	↓	Atg5 depletion in tumor ECs	Exacerbation of functional abnormalities in tumor vasculature	[135]
	↓	3MA inhibition of mitophagy in retinal ECs	Improved cell viability, reduced tube formation and cell migration	[136]

**Table 2 cells-13-00825-t002:** Proposed therapeutic modulators of autophagy—some of the known pharmacologic and non-pharmacologic agents that have been implicated in the process of autophagy.

Compound	Type of Autophagy Regulation	Stage of Clinical Development	Ref
**Pomegranate**	Upregulation of mitophagy via TFEB activation	Preclinical	[146]
***Ginkgo biloba* extract**	Upregulation of FUNDC1-dependent mitophagy	Preclinical	[147]
**PMI (p62-mediated mitophagy inducer)**	Upregulation of mitophagy	Preclinical	[148]
**Quercetin**	Upregulation of autophagy in endothelial cells	Preclinical	[149]
**AUTAC (autophagy-targeting chimera)**	Upregulation of mitophagy	Preclinical	[150]
** *mTORC inhibitors* ** **Rapamycin** **Everolimus** **Temsirolimus** **Ridaforolimus** **Deforlimus**	Upregulation of mitophagy via p62/Parkin Upregulation of autophagy via mTOR inhibition	Approved for other indications (rapamycin, everolimus) Preclinical (temsirolimus, ridaforolimus, deforolimus)	[151,152,153,154]
** *SIRT1 activators* ** **Resveratrol** **Polydatin** **Salidroside**	Upregulation of autophagy through activation of TFEB Upregulation of autophagy through enhancement of the SIRT1 and FOXO pathways	Approved for other indications	[155,156,157]
**Melatonin**	Upregulation of mitophagy through activation of SIRT3-FOXO3–Parkin Potential downregulation or prevention of excessive mitophagy	Approved for other indications	[130,158]
**Metformin**	Upregulation of autophagy and mitophagy via AMPK activation as well as PINK1, Parkin, Nix, and LC3 upregulation	Approved for other indications	[159]
**Statins**	Upregulation of autophagy via mTOR inhibition	Approved for other indications	[160,161,162,163]
**Exercise**	Upregulation of autophagy	Approved for other indications	[164,165,166,167]
**Caloric restriction**	Upregulation of autophagy	Approved for other indications	[168,169]
**UPS30 inhibitors**	Downregulation of mitophagy via inhibition of Parkin-mediated mitophagy	Preclinical	[170]

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
