# Peer review of "The Role of Autophagy in Vascular Endothelial Cell Health and Physiology"

_cells, 2024, doi:10.3390/cells13100825_

Round 1

Reviewer 1 Report

Comments and Suggestions for Authors

Hu’s manuscript reviews the roles of autophagy in endothelial cells and its implications in various diseases. Endothelial cells play critical roles in regulating vascular tone, permeability, angiogenesis, and tissue hemostasis. Endothelial dysfunction has been linked to a range of diseases including atherosclerosis and inflammation. Autophagy serves as an effective mechanism to maintain cellular hemostasis by degrading organelles and proteins. Therefore, autophagy plays essential roles in endothelial cells and its dysregulation can lead to various diseases. to make the review more accessible and informative for readers, some revisions are suggested.

1.     In the section “molecular mechanisms of autophagy”, autophagy should be introduced by discussing the key molecules and signaling pathways. Similar discussion is suggested for the following sections as well.

2.     Regarding the figure 1, the current version appears to be quiet basic. Key molecules involved in autophagy should be incorporated. Please also illustrate different types of autophagy, in order to provide a more comprehensive understanding of the process.

3.     The necessity of Figure 2 is low. Its contribution to the review is not significant.

4.     A revision is required for Table 1. Although pharmacological modulators of autophagy are listed, it is important to note that none of these modulators have been specifically utilized in the context of endothelial cells or their related diseases. Instead of the current pharmacological modulators, it is recommended to include a new list that focuses on key molecules involved in autophagy. This list should outline their functions in endothelial cells and vascular diseases, as well as the study methods (such as knockdown or knockout) used to investigate their roles. This revision will provide a more comprehensive and relevant overview of autophagy in the context of endothelial cells and associated diseases.

Author Response

Reviewer#1

Q1. In the section “molecular mechanisms of autophagy”, autophagy should be introduced by discussing the key molecules and signaling pathways. Similar discussion is suggested for the following sections as well.

Answer. We thank the reviewer for these suggestions. We have changed the introduction, including an overview of the key molecules and important signaling pathways.

Q2. Regarding the figure 1, the current version appears to be quiet basic. Key molecules involved in autophagy should be incorporated. Please also illustrate different types of autophagy, in order to provide a more comprehensive understanding of the process.

Answer. We are thankful for referee’s careful evaluation of this manuscript.  We have revised Figure 1 to show the process of autophagy in greater detail, including key molecules. As this review mainly highlights macroautophagy and selective mitophagy, we have revised the figure to illustrate the key differences between these two forms of autophagy.

Q3. The necessity of Figure 2 is low. Its contribution to the review is not significant.

Answer. Figure 2 was removed.

Q4.  A revision is required for Table 1. Although pharmacological modulators of autophagy are listed, it is important to note that none of these modulators have been specifically utilized in the context of endothelial cells or their related diseases. Instead of the current pharmacological modulators, it is recommended to include a new list that focuses on key molecules involved in autophagy. This list should outline their functions in endothelial cells and vascular diseases, as well as the study methods (such as knockdown or knockout) used to investigate their roles. This revision will provide a more comprehensive and relevant overview of autophagy in the context of endothelial cells and associated diseases.

Answer. We have replaced the previous Table 1 with a table summarizing select findings on the role of autophagy in vascular diseases, including the study methods and the observed effects. Although none of the pharmacological modulators have been specifically utilized in the context of ECs for their specific diseases, we felt it would be helpful for readers to have a visual aid summarizing the state of the research on autophagy modulators so far, as they are a promising future direction for the treatment of vascular diseases. As such, that table has now become Table 2.

Reviewer 2 Report

Comments and Suggestions for Authors

In this manuscript, Hu et al. review the role of autophagy in vascular endothelial cells (ECs) pathophysiology. In addition, they summarize the compounds that modulate autophagy in ECs for their clinical application.

Comments:

Major

-Lipophagy is not specifically mentioned in the manuscript. Lipophagy plays a pivotal role in EC injury.

-Epigenetic regulation of autophagy in vascular endothelial cells is another interesting topic which is not addressed in the manuscript.

-Therapeutic approaches: Please, include in the text and table 1: exercise, caloric restriction, and statins.

Minor

Table 1 needs references

Author Response

Reviewer#2

Q1. Lipophagy is not specifically mentioned in the manuscript. Lipophagy plays a pivotal role in EC injury.

Answer. We thank the reviewer for these suggestions – we have added additional information on the role of lipophagy in the pathogenesis of atherosclerosis, as well as an additional section on the epigenetic regulation of autophagy in vascular ECs.

Q2. Epigenetic regulation of autophagy in vascular endothelial cells is another interesting topic which is not addressed in the manuscript.

Answer. We have modified manuscript based on reviewer’s comments.

Q3.Therapeutic approaches: Please, include in the text and table 1: exercise, caloric restriction, and statins.

Answer. We have added sections in the text and in Table 2 (previously Table 1) on exercise, caloric restriction, and statins.

Q4. needs references.

Answer. References added to table.

Round 2

Reviewer 2 Report

Comments and Suggestions for Authors

The manuscript was improved. However, given the extensive information provided, I suggest incorporating more figures summarizing the contents of the different sections.

Author Response

We thank the reviewer for the suggestion. A new figure 2 has been added to the revised manuscript. 

Round 3

Reviewer 2 Report

Comments and Suggestions for Authors

The authors have addressed all my concerns